# Accelerated Burn Healing in a Mouse Experimental Model Using α-Gal Nanoparticles

**DOI:** 10.3390/bioengineering10101165

**Published:** 2023-10-06

**Authors:** Uri Galili

**Affiliations:** Department of Medicine, Rush University Medical College, Chicago, IL 60612, USA; uri.galili@rcn.com; Tel.: +1-312-753-5997

**Keywords:** burn healing, anti-Gal antibody, α-gal epitope, α-gal nanoparticles, macrophage migration, α-gal therapy

## Abstract

Macrophages play a pivotal role in the process of healing burns. One of the major risks in the course of burn healing, in the absence of regenerating epidermis, is infections, which greatly contribute to morbidity and mortality in such patients. Therefore, it is widely agreed that accelerating the recruitment of macrophages into burns may contribute to faster regeneration of the epidermis, thus decreasing the risk of infections. This review describes a unique method for the rapid recruitment of macrophages into burns and the activation of these macrophages to mediate accelerated regrowth of the epidermis and healing of burns. The method is based on the application of bio-degradable “α-gal” nanoparticles to burns. These nanoparticles present multiple α-gal epitopes (Galα1-3Galβ1-4GlcNAc-R), which bind the abundant natural anti-Gal antibody that constitutes ~1% of immunoglobulins in humans. Anti-Gal/α-gal nanoparticle interaction activates the complement system, resulting in localized production of the complement cleavage peptides C5a and C3a, which are highly effective chemotactic factors for monocyte-derived macrophages. The macrophages recruited into the α-gal nanoparticle-treated burns are activated following interaction between the Fc portion of anti-Gal coating the nanoparticles and the multiple Fc receptors on macrophage cell membranes. The activated macrophages secrete a variety of cytokines/growth factors that accelerate the regrowth of the epidermis and regeneration of the injured skin, thereby cutting the healing time by half. Studies on the healing of thermal injuries in the skin of anti-Gal-producing mice demonstrated a much faster recruitment of macrophages into burns treated with α-gal nanoparticles than in control burns treated with saline and healing of the burns within 6 days, whereas healing of control burns took ~12 days. α-Gal nanoparticles are non-toxic and do not cause chronic granulomas. These findings suggest that α-gal nanoparticles treatment may harness anti-Gal for inducing similar accelerated burn healing effects also in humans.

## 1. Introduction

Macrophages play a pivotal role in the process of wound and burn healing [1,2,3,4]. In both types of healing, the M1 macrophages first debride the injured skin of apoptotic and dead cells and of the intercellular matrix. Subsequently, M2 macrophages orchestrate the regeneration of the epidermis, dermis, and hypodermis of the injured skin [2,3,4,5,6]. This is mediated by a wide range of cytokines/growth factors secreted by these macrophages, including vascular endothelial growth factor (VEGF) mediating neo-vascularization, epidermal growth factor (EGF) inducing epidermal regrowth, fibroblasts growth factor (FGF) recruiting fibroblasts, and factors recruiting mesenchymal stem cells (MSCs), which contribute to the regeneration of the injured skin [7]. In wounds, incisions, and contusions, the macrophages mediating healing comprise both residential macrophages and monocyte-derived macrophages that are recruited by chemotactic factors such as macrophage inflammatory protein-1 (MIP-1), monocyte chemoattractant protein-1 (MCP-1) regulated on activation, and normal T cell expressed and secreted factor (RANTES) secreted by cells surrounding the wound [8,9,10,11]. However, since in epidermis-penetrating burns (burn degrees 2–4), the residential macrophages are inactivated or killed [1], and since the surface area size of the burns is in many cases larger than that of wounds, infiltration of macrophages into burns and healing of burns may take longer time than in wounds, and the regeneration of the epidermis in many burns may be slower than in wounds. This slow re-epithelialization is a major risk factor because of microbial infections that occur in the absence of intact epidermis. Such infections may result in high morbidity and mortality following severe burn injuries [6,12,13,14].

Based on the pivotal role of macrophages in the healing of burns, and in view of the immune suppression of macrophages following burn injury [15,16], it has been suggested that the risk factors due to slow re-epithelialization might be reduced by accelerating the regrowth of the epidermis over the burned tissue [3,4,7,15,16,17,18]. Several methods of various degrees of difficulty have been studied for accelerating burn healing. These include topical application to burns of autologous MSCs [19,20], autologous cultured epidermal cell grafts [21,22], recombinant human granulocyte–macrophage colony-stimulating factor (GM-CSF) [23,24], high-density lipoprotein nanoparticles [25], bioactive molecules delivered in microfibers [26,27], and the use of negative pressure wound therapy [28].

The present review offers an alternative method to those mentioned above, supporting the accelerated healing of burns by inducing the rapid recruitment and activation of macrophages in treated burns by topical application of α-gal nanoparticles [17]. This method recapitulates the physiologic healing processes of burns, but the accelerated recruitment of macrophages into treated burns cuts the healing time by half. The interaction of these nanoparticles with the natural anti-Gal antibody (one of the most abundant natural antibodies in humans) within burns results in rapid and extensive recruitment of monocyte-derived macrophages into burns [17,18]. Many of these macrophages polarize into M2 macrophages, which orchestrate the accelerated healing of burns by the localized secretion of angiogenic factors such as VEGF and growth factors recruiting MSCs. This review describes studies that characterized the anti-Gal antibody and α-gal nanoparticles, the simple production of these nanoparticles from rabbit red blood cells, and the great efficacy of the burn and wound therapies with α-gal nanoparticles as observed in the anti-Gal-producing mouse experimental model.

## 2. Anti-Gal and the α-Gal Epitope

The method described in this review for accelerating burn healing harnesses the immunologic potential of the natural anti-Gal antibody, which is one of the most abundant natural antibodies in humans, constituting 1% of serum immunoglobulins [29]. The immune system in humans produces anti-Gal throughout life in response to antigenic stimulation by some carbohydrate antigens presented on gastrointestinal bacteria [30,31]. The mammalian antigen recognized by anti-Gal is the α-gal epitope (Galα1-3Galβ1-4GlcNAc-R) [32,33,34]. The α-gal epitope is abundantly expressed on glycolipids and glycoproteins of non-primate mammals, lemurs, and New World monkeys (monkeys of South America); therefore, these mammals cannot produce anti-Gal [35,36,37]. In contrast, humans, apes, and Old World monkeys (monkeys of Asia and Africa) all lack α-gal epitopes but produce the natural anti-Gal antibody [35,36,37,38]. Incubation of cells presenting α-gal epitopes in human serum results in effective activation of the complement cascade in the serum because of the binding of serum anti-Gal to these α-gal epitopes. The efficacy of this complement activation was demonstrated in xenotransplantation studies. Interaction between anti-Gal and α-gal epitopes on endothelial cells of pig xenografts was found to result in the activation of the complement system, causing cytolysis of these cells, the collapse of the vascular bed, and rapid (hyperacute) rejection of such xenografts in monkeys or humans [39,40,41,42]. Similarly, incubation of enveloped viruses presenting α-gal epitopes in human serum was found to result in binding of anti-Gal to these epitopes and activation of the complement system, which led to complement-mediated destruction of such viruses [43,44,45,46]. Since the very potent macrophage-directing chemotactic factors C5a and C3a are produced as complement cleavage peptide byproducts during complement activation, we assumed that binding of serum anti-Gal to multiple α-gal epitopes on α-gal nanoparticles applied to burns and wounds may result in extensive recruitment of monocyte-derived macrophages to treated skin injury sites [17,47,48].

## 3. Hypothesis

The effective complement activation by anti-Gal binding to α-gal epitopes led us to hypothesize that topical application of nanoparticles presenting multiple α-gal epitopes (called α-gal nanoparticles and previously called α-gal liposomes) during the early stages of hemostasis in burns results in the binding of the natural anti-Gal antibody to these nanoparticles [17,47]. Anti-Gal is present in the fluid film on the surface of burns together with the complement system proteins, as well as with other serum proteins that leak from injured capillaries. As detailed below, α-gal nanoparticles are small-size liposomes (~100–300 nm) constructed from α-gal-presenting glycolipids that are anchored in the nanoparticle wall that is composed of phospholipids and cholesterol (Figure 1A) [17,47].

The binding of anti-Gal to the multiple α-gal epitopes on the nanoparticles results in activation of the complement cascade and thus in generation of chemotactic complement cleavage peptides C5a and C3a (Step 1 in Figure 1B) [17,47]. These chemotactic factors induce extensive migration of neutrophils and monocyte-derived macrophages into the burn area (Step 2 in Figure 1B). In addition, it was hypothesized that, whereas the neutrophils survive only for a few hours in the burn, the recruited macrophages are long-lived and that they bind the α-gal nanoparticles as a result of the interaction between the Fcγ “tail” of anti-Gal bound to the nanoparticles and Fcγ receptors (FcγR) on the macrophages (Step 3 in Figure 1B). It was further hypothesized that the multiple Fcγ/FcγR interactions between anti-Gal-coated α-gal nanoparticles and macrophages may activate the recruited macrophages to secrete various cytokines/growth factors that mediate accelerated migration of fibroblasts and MSCs into the treated burn, as well as neo-vascularization of the healing burn (Step 4 in Figure 1B) and rapid re-epithelialization. Ultimately, these multiple cytokines/growth factors secreted by the recruited and activated macrophages may accelerate the healing of the α-gal nanoparticles-treated burns in comparison to non-treated burns.

A relatively simple way of preparing α-gal nanoparticles is by extraction of their components from the membranes (ghosts) of rabbit red blood cells (RBCs) in a mixture of chloroform and methanol [17,48]. The reason for using these RBCs is that they present as many as 2 × 10^6^ α-gal epitopes per RBC, an amount that is several folds higher than any other mammalian RBC studied [35,47]. After the rabbit RBCs are lysed in water and their membranes are washed for the removal of hemoglobin, the RBC membranes are incubated overnight in a solution of chloroform:methanol 1:2 with constant stirring, resulting in the extraction of phospholipids, glycolipids, and cholesterol from these membranes, whereas all proteins are denatured and removed from the solution by filtration [17,48]. The solution containing the extracted molecules is dried and the mixture of phospholipids, glycolipids, and cholesterol is resuspended in saline by extensive sonication. This sonication results in the formation of a suspension of submicroscopic liposomes (~100–300 nm) with walls comprising phospholipid and cholesterol and studded with multiple α-gal epitopes in the form of anchored α-gal glycolipids (Figure 1A) [47]. These submicroscopic liposomes originally called α-gal liposomes [17,48] have been subsequently referred to as α-gal nanoparticles [18,47] to indicate that they do not contain any substance in their lumen.

The α-gal nanoparticles were found to present ~10^14^ α-gal epitopes/mg nanoparticles [18]. Processed 500 mL of packed rabbit RBCs were found to yield ~6 gm of α-gal nanoparticles. Because of their small size, α-gal nanoparticle suspensions can be sterilized by filtration through a 0.4 μm filter [17,48]. It is of note that α-gal nanoparticles may be prepared also from synthetic α-gal glycolipids, phospholipids, and cholesterol by similar mixing and sonication processes. The α-gal nanoparticles are highly stable and can be kept as frozen suspensions or at 4 °C for >4 years and as dried nanoparticles on wound dressings kept at room temp. for >1 year [47]. This stability of the stored α-gal nanoparticles could be confirmed by their ability to bind anti-Gal in amounts like those measured immediately after production. Topical application of α-gal nanoparticles to burns and wounds can be performed by using various methods including the use of nanoparticle suspensions in saline or PBS, nanoparticles dried on wound dressings, aerosol suspensions, and suspensions in hydrogels [18].

## 4. Experimental Animal Models

Studies of anti-Gal-associated therapies cannot be performed in standard animal experimental models such as mice, rats, rabbits, pigs, or guinea pigs because these animals, like other non-primate mammals, synthesize α-gal epitopes [35,36]. Therefore, such mammals are immunotolerant to the α-gal epitope and cannot produce the anti-Gal antibody [35]. However, the two experimental non-primate mammalian models available for studying anti-Gal-associated therapies have been mice [49,50] and pigs [51,52] in which the *GGTA1* gene coding for the glycosyltransferase synthesizing α-gal epitope “α1,3galactosyltransferase” is disrupted (i.e., knocked out). These α1,3galactosyltransferase knockout (GT-KO) mice [17,18] and pigs [53,54,55] lack the ability to synthesize α-gal epitopes and thus can produce the anti-Gal antibody. Whereas GT-KO pigs produce the natural anti-Gal antibody, as humans do, GT-KO mice do not naturally produce this antibody since they do not develop gastrointestinal bacterial flora that may immunize them because they are kept in a sterile environment and receive sterile food. Nevertheless, GT-KO mice readily produce the anti-Gal antibody following several immunizations with xenogeneic cells or tissues presenting α-gal epitopes, such as pig kidney membranes (PKM) homogenate [56].

## 5. In Vitro Effects of α-Gal Nanoparticles on Macrophages

Some of the steps of anti-Gal/α-gal nanoparticle interaction, described in the hypothesis illustrated in Figure 1B [47], could be demonstrated in in vitro studies described in Figure 2. Step 1 of anti-Gal binding to α-gal epitopes on α-gal nanoparticles was demonstrated by the specific binding of monoclonal anti-Gal antibody to these nanoparticles (Figure 2A) [17,47]. A similar binding was observed with serum anti-Gal from anti-Gal-producing GT-KO mice that interacts with α-gal nanoparticles (Figure 2B). In the absence of α-gal epitopes on the nanoparticles, no binding of the monoclonal anti-Gal antibody was observed [17].

Step 3 in the hypothesis in Figure 1B predicts the binding of anti-Gal-coated α-gal nanoparticles to macrophages via Fcγ/FcγR interaction. This binding is demonstrated through scanning electron microscopy (SEM) in Figure 2C,D [47]. Macrophages lacking α-gal epitopes were generated in vitro by the culturing of monocytes obtained from the blood of GT-KO pigs. These macrophages were co-incubated for 2 h at room temp. with anti-Gal-coated α-gal nanoparticles. Such incubation resulted in the extensive binding of α-gal nanoparticles to the macrophages, shown as the multiple small spheres (size of 100–300 nm) covering the surface of the two macrophages (Figure 2C,D). In the absence of anti-Gal on the α-gal nanoparticles, no binding of these nanoparticles to macrophages was observed [47]. Step 4 in the hypothesis in Figure 1B predicted that the binding of anti-Gal-coated α-gal nanoparticles to recruited macrophages via Fcγ/FcγR interaction may generate signals that activate a variety of cytokines/growth factors producing genes that orchestrate the accelerated healing of α-gal nanoparticle-treated burns. The possible activation of macrophages by anti-Gal-coated α-gal nanoparticles was studied with GT-KO mouse macrophages incubated for 24–48 h at 37 °C, alone or with α-gal nanoparticles coated with anti-Gal or lacking the antibody. The secretion of VEGF by the macrophages was measured in the tissue culture medium after 24 and 48 h of co-incubation. Macrophages co-incubated with α-gal nanoparticles lacking anti-Gal secreted only a background level of VEGF, as determined by using ELISA measuring this cytokine (Figure 2E). However, co-incubation of the macrophages with anti-Gal-coated α-gal nanoparticles for 24 and 48 h resulted in elevated secretion of VEGF by the activated macrophages at levels that were significantly higher than the background levels (Figure 2E) [47,48]. These findings indicated that anti-Gal-mediated binding of α-gal nanoparticles to cultured macrophages indeed induces these macrophages to secrete VEGF.

## 6. In Vivo Effects of α-Gal Nanoparticles on Macrophages

A crucial step in the hypothesis in Figure 1B is Step 2, which predicts that anti-Gal binding to α-gal nanoparticles applied to burns activates the complement system, resulting in the formation of complement cleavage chemotactic peptides C5a and C3a. These peptides direct a rapid and extensive recruitment of monocyte-derived macrophages to the treated burn. The occurrence of Step 2 was studied through intradermal injection of 10 mg of α-gal nanoparticles in anti-Gal-producing GT-KO mice (i.e., GT-KO mice immunized with PKM) and microscopic evaluation of macrophages in the injection site at various time points. The first effect of such an injection was the accumulation of many neutrophils, observed at the injection site within 12 h post-injection [48]. These neutrophils are also chemotactically recruited by C5a and C3a generated by the anti-Gal/α-gal nanoparticle interaction. However, after 24 h, most neutrophils disappeared, and multiple mononuclear cells were observed migrating to the injection site (Figure 3A) [47,48]. The number of macrophages increased after 4 days as expected, and they all were immunostained by the macrophage-specific antibody F4/80 (Figure 3B). Quantitative real-time PCR (qRT-PCR) of a skin specimen containing the recruited macrophages displayed activation of genes encoding for fibroblast growth factor (FGF), interleukin 1 (IL1), platelet-derived growth factor (PDGF), and colony-stimulating factor (CSF) [48].

The number of recruited macrophages further increased by day 7 (Figure 3C). These macrophages had a large size and ample cytoplasm, characteristic of activated macrophages (Figure 3C,D) [48]. Large numbers of recruited macrophages were observed at the injection site, even on day 14. However, by day 21 post-injection of the α-gal nanoparticles, all macrophages disappeared from the injection site, and the skin in that area displayed a normal structure with no granuloma, chronic inflammatory response, or keloid formation [48]. Intradermal injection of α-gal nanoparticles together with the cobra venom factor (a complement activation inhibitor), saline, or nanoparticles lacking α-gal epitopes (i.e., nanoparticles produced from GT-KO pig RBC) all resulted in no significant recruitment of macrophages to the injection site [48]. Similarly, intradermal injection of α-gal nanoparticles into wild-type (WT) mice (i.e., mice lacking the anti-Gal antibody) resulted in no macrophage recruitment. These observations clearly demonstrate the ability of α-gal nanoparticles to induce extensive and rapid recruitment of macrophages by the binding of the anti-Gal antibody and activation of the complement system, which generates the potent chemotactic complement cleavage peptides C5a and C3a [48].

## 7. Macrophages Recruited by α-Gal Nanoparticles Are M2 Further Recruiting MSCs

Analysis of the characteristics of macrophages recruited by α-gal nanoparticles in anti-Gal-producing GT-KO mice could be further performed by the subcutaneous implantation of biologically inert sponge discs (made of polyvinyl alcohol- PVA, 10 mm diameter, 3 mm thickness) that contained 10 mg α-gal nanoparticles. The PVA sponge discs were explanted on day 6 or day 9. The cells harvested from these sponge discs had the morphology of large macrophages like those presented in Figure 3D. Each of the PVA sponge discs contained, at those time points, ~0.4 × 10^6^ and ~0.6 × 10^6^ infiltrating cells, respectively, whereas sponge discs with only saline contained ~0.02 × 10^6^ and ~0.04 × 10^6^ cells, respectively [17]. Immunostaining and flow cytometry analysis of the cells recruited by α-gal nanoparticles indicated that most of them (>90%) expressed the macrophage markers CD11b and CD14 (Figure 4A). In contrast, no significant proportion of the infiltrating cells displayed surface markers of CD4+ T cells, CD8+ T cells, or B cells (i.e., lymphocytes presenting CD20+ cell marker) [17,57].

Further analysis of the polarization state of macrophages recruited by α-gal nanoparticles indicated that they were M2 macrophages since they were positively immunostained for M2 markers IL-10 and Arginase-1 and were negatively immunostained for IL-12, a marker that characterizes M1 macrophages (Figure 4B) [57,58]. When these infiltrating macrophages were cultured in vitro for 5 days, the culture wells were found to contain cell colonies at a frequency of 1 colony per 50,000 to 100,000 cultured macrophages. These colonies had the morphological characteristics of colonies formed by MSCs (Figure 4C,D) [47,57]. Accordingly, the majority of the cells retrieved from these colonies presented the stem cell markers Sca-1 and CD-29 (Figure 4E,F). These colonies contained 300–1000 cells per colony, suggesting that the cells forming them proliferated at an average cell cycle time of ~12 h. Overall, the observations in Figure 4B–F suggest that most macrophages recruited and activated by α-gal nanoparticles polarized into M2 macrophages and further directed the migration of MSCs into the implanted PVA sponge discs [57].

## 8. Accelerated Healing of Burns by Topical Application of α-Gal Nanoparticles

The above in vitro and in vivo studies on the effects of α-gal nanoparticles on macrophages (Figure 3 and Figure 4, respectively) prompted the analysis of the effects of these nanoparticles on burns healing in anti-Gal-producing GT-KO mice [17]. For this purpose, 10 mg of α-gal nanoparticles from a suspension containing 100 mg/mL was dried under sterile conditions on 1 × 1 cm pads of small “spot” bandages. Pads with dried 0.1ml saline served as controls. Two thermal injuries were performed on two shaved abdominal flanks of anesthetized mice by a brief contact with the heated end of a metal spatula (~2 × 3 mm), resulting in a second-degree burn affecting the epidermis and dermis but not the hypodermis. The right-side burns were covered with α-gal nanoparticle-coated spot bandages, and the left-side burns were covered with control spot bandages containing dried saline (Figure 5A). Removal of the bandages by the mice was prevented by covering them with Tegaderm^TM^ and Transpore^TM^ adhesive tape. The dressings were removed at various time points, the extent of covering the burn by re-epithelialization and macrophage infiltration was measured, and the burn areas were sectioned and subjected to histologic staining using H&E (Figure 5B,C and Figure 6) [17] and Mason trichrome that stains collagen blue (Figure 7) [17]. The extent of macrophage infiltration into burns and re-epithelialization (i.e., covering of the burn injury by the regenerating epidermis) is presented in Figure 8A,B, respectively [17].

The histology of a representative normal mouse skin is presented in Figure 6A and Figure 7F. The epidermis in such skins comprises 2–3 layers of epithelial cells, the underlying dermis is stained pink by H&E and blue by Mason trichrome. The hypodermis contains mostly fat tissue characterized by multiple adipocytes. The thermal injuries in the mouse skin resulted in the destruction of both the epidermis and the dermis, as observed 24 h post-injury (Figure 6B). This damage is similar to second-degree burns in humans in that both the epidermis and dermis are destroyed, whereas damage to the hypodermis is minimal. No differences were observed 24 h post-injury in α-gal nanoparticle-treated burns (Figure 6B) and in burns treated with saline 17.

A major difference was observed between treated and control burns, inspected 3 days post-injury. Whereas no significant number of macrophages was observed in control burns, as many as 40 macrophages were detected in the same size field in α-gal nanoparticle-treated burns (Figure 6C,D and Figure 8A). In addition, control burns displayed some degree of neutrophil infiltration, but α-gal nanoparticle-treated burns displayed a ~5-fold higher number of neutrophils (Figure 6C,D).

The most dramatic difference between the two burn treatments was observed 6 days post thermal injury. At that time point, α-gal nanoparticle-treated burns displayed extensive regeneration of the epidermis as 50–100% re-epithelialization of the surface areas (mean of ~70%) (Figure 8B). The newly formed epidermis also included the formation of *stratum corneum* (Figure 5C and Figure 6F). Many of the macrophages and neutrophils were found to be removed to the surface of the regenerating epidermis, within and above the *stratum corneum,* and were mixed with remnants of the eschar. This accelerated healing was found to be dose-dependent since 1 mg of α-gal nanoparticles induced an average of 23% healing after 6 days, and 0.1 mg elicited no measurable healing [17]. No significant epidermis regeneration was observed on day 6 in control burns (Figure 5B, Figure 6E and Figure 8B). Nevertheless, the dermis displayed increasing numbers of macrophages in a state of migration to the apical area of the injured dermis.

Evaluation of dermis regeneration was performed by using Mason trichrome, which stained blue de novo synthesized collagen. Near-complete regeneration of the dermis was observed in α-gal nanoparticle-treated burns after 6 days (Figure 7D) [17], whereas in control burns, much of the dermis was stained red, characteristic of thermal damage of the dermis (Figure 7A,C). An initial indication of the re-epithelialization of control burns was observed on day 9, where ~20% of the burn surfaces were covered by the regenerating epidermis (Figure 8B) [17]. In contrast, 100% of the α-gal nanoparticle-treated burns were healed by that time point. By day 12, all control burns displayed complete healing as that observed in α-gal nanoparticle-treated burns (Figure 6G,H, Figure 7E, and Figure 8B). These findings imply that topical application of α-gal nanoparticles to burns accelerates burn healing and cuts the healing time by ~40–50%. It is of note that in the absence of anti-Gal (e.g., in wild-type mice), no difference in the healing process was observed between control burns and burns treated with α-gal nanoparticles. Both were similar to the healing of control burns in anti-Gal-producing GT-KO mice [17].

## 9. α-Gal Nanoparticles Mediated Accelerated Healing of Wounds

Since both healing processes of burns and wounds are mediated by macrophages, it was of interest to determine whether the α-gal nanoparticle treatment has accelerating effects on wound healing in anti-Gal-producing GT-KO mice, similar to the effects described above in burn healing. Oval-shaped full-thickness wounds (~6 × 9 mm) were made in anesthetized anti-Gal-producing GT-KO mice. The wounds were covered with spot bandage dressings containing 10 mg of dried α-gal nanoparticles or with control spot bandage dressings containing dried saline. The healing of the wounds was evaluated by re-epithelialization at various time points. As with treated burns, wounds treated with α-gal nanoparticles completely healed by day 6 post-treatment, whereas control wounds healed only after 12–14 days [18,48]. Studies on completely healed treated and control wounds 28 days post-injury indicated that healing of saline-treated control wounds resulted in fibrosis and scar formation, characteristic of the physiologic default healing of untreated wounds. In contrast, healing of α-gal nanoparticle-treated wounds resulted in the restoration of the normal structure of the skin, including the re-appearance of skin appendages such as hair, sebaceous glands, and hypodermal adipocytes [48]. It was suggested that the accelerated recruitment and activation of macrophages resulted in the regeneration of the normal skin structure prior to the initiation of the default fibrosis and scar formation processes, thereby avoiding the latter processes [18,47]. The accelerated wound healing by these α-gal nanoparticles was further validated by an independent laboratory in anti-Gal-producing GT-KO healthy mice [59], diabetic mice [58], and mice following skin irradiation [60]. It should be noted that similar healing that included the restoration of the original structure and function was observed in anti-Gal-producing GT-KO mice following myocardial infarction (MI) and treatment by injections of α-gal nanoparticles [61]. In contrast, post-MI ischemic myocardium injected with saline displayed fibrosis and scar formation, similar to the pathology observed in post-MI injured myocardium in humans.

## 10. Concluding Remarks

Burn healing can be accelerated by the use of α-gal nanoparticles, which harness the immunologic potential of the natural anti-Gal antibody, an abundant antibody in humans constituting ~1% of immunoglobulins. Application of α-gal nanoparticles to burns results in the binding of anti-Gal to the α-gal epitopes on these nanoparticles. This interaction activates the complement system, resulting in the formation of complement cleavage chemotactic peptides, which direct rapid and extensive migration of monocyte-derived macrophages into the treated burns. These recruited macrophages bind via their Fcγ receptors the Fcγ “tails” of anti-Gal coating the α-gal nanoparticles and are activated into an M2 polarization state. The activated macrophages further produce a variety of cytokines/growth factors that mediate accelerated regrowth of the epidermis and regeneration of the injured dermis. In anti-Gal-producing mice, the accelerated epidermal regrowth results in the covering of the burn with an intact epidermis twice as fast as the physiologic regrowth. Similarly, the healing of α-gal nanoparticle-treated burns in these mice is 40–60% faster than physiologic burn healing. The α-gal nanoparticles are non-toxic and do not induce chronic granulomas. In addition, the α-gal nanoparticles are highly stable for long periods at various temperatures. In view of their accelerated healing effects, α-gal nanoparticles may be considered for the treatment of human burns. Accelerated healing using α-gal nanoparticles is also observed in treated wounds of anti-Gal-producing mice. Application of α-gal nanoparticles to burns and wounds may be feasible in the form of dried nanoparticles on wound dressings and as suspensions, aerosols, and hydrogels or incorporated into sheets of biodegradable scaffold materials such as collagen sheets.

## Figures and Tables

**Figure 1 bioengineering-10-01165-f001:**
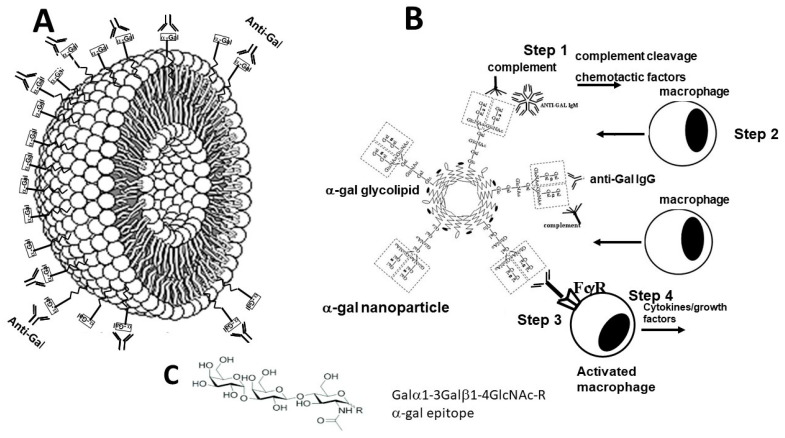
Illustration of an α-gal nanoparticle (**A**), the hypothesized immune processes induced by α-gal nanoparticles applied to burns (**B**), and the structure of the α-gal epitope (**C**). (**A**) The α-gal nano-particles present multiple α-gal epitopes (rectangles) on glycolipids which are anchored in the phospholipid bilayer that forms the wall of the nanoparticle. The nanoparticle wall may also contain cholesterol, which stabilizes the wall. The natural anti-Gal antibody readily binds to these α-gal epitopes (structure illustrated in (**C**)). (**B**) The steps hypothesized to occur in burns after application of α-gal nanoparticles: Step 1—anti-Gal binding to α-gal nanoparticles activates the complement system. Step 2—the complement cleavage peptides C5a and C3a formed as a result of complement activation function as chemotactic factors that direct extensive and rapid recruitment of mono-cyte-derived macrophages into the treated burns. Step 3—the recruited macrophages interact via their Fcγ receptors (FcγR) with the Fcγ portion (tail) of anti-Gal coating the α-gal nanoparticles. Step 4—the Fcγ/FcγR interactions activate the macrophages to secrete cytokines/growth factors that induce accelerated healing of the treated burns. (**C**) Detailed structure of the α-gal epitope, which is constructed of galactose (Gal) linked α1,3 to a penultimate galactose (Gal) that is linked β1–4 to an N-acetylglucosamine (GlcNAc). Adapted with permission from [47]. 2018, Elsevier.

**Figure 2 bioengineering-10-01165-f002:**
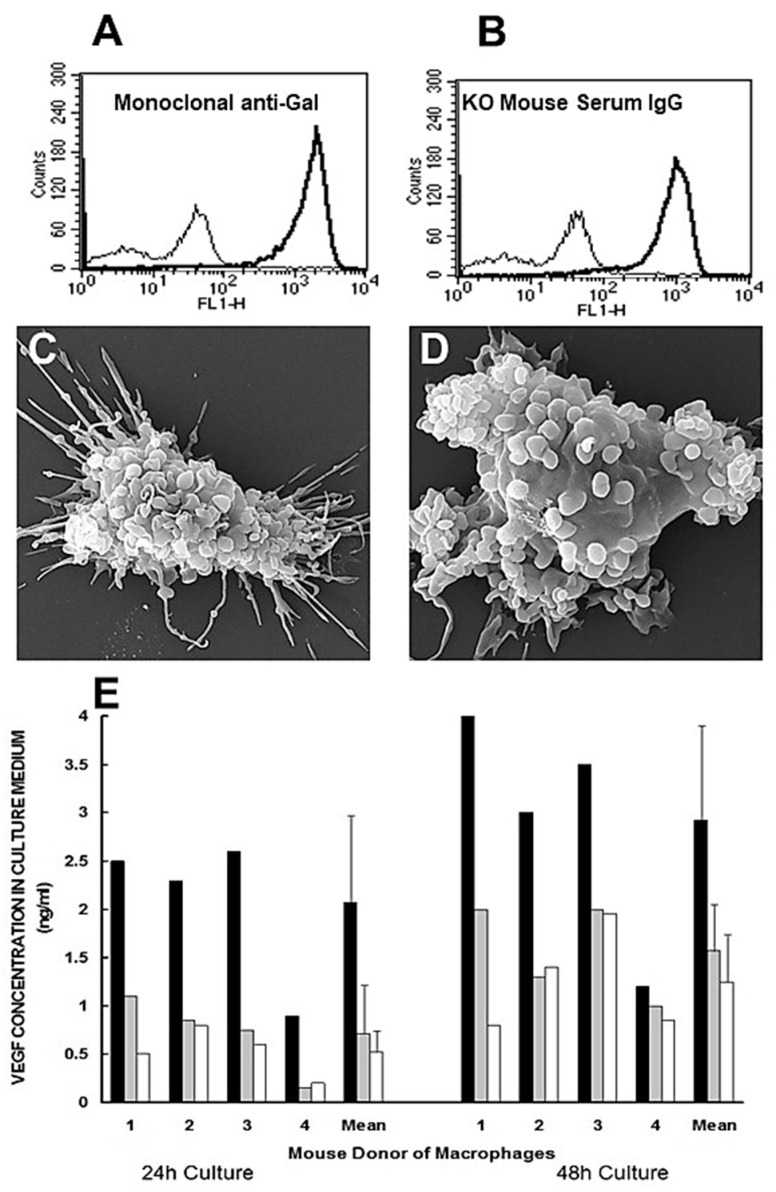
In vitro demonstration of anti-Gal binding to α-gal nanoparticles and the resulting effects on macrophages. (**A**) Binding of monoclonal anti-Gal IgM antibody to α-gal epitopes on α-gal nanoparticles. The thin line represents the IgM isotype control. (**B**) As in (**A**), using anti-Gal IgG in serum of α1,3galactosyltransferase knockout (GT-KO) mouse producing anti-Gal. The thin line represents the IgG isotype control. (**C**,**D**) Binding of anti-Gal-coated α-gal nanoparticles to adherent GT-KO pig macrophages as shown by scanning electron microscopy (SEM) after 2 h incubation of anti-Gal-coated nanoparticles with the macrophages at room temp. followed by washings to remove nonadherent nanoparticles. The surfaces of representative macrophages are covered with α-gal nanoparticles that have the shape of small spheres. The size of the α-gal nanoparticles is ∼100–300 nm. (**E**) GT-KO mouse peritoneal macrophage secretion of VEGF following incubation with anti-Gal-coated α-gal nanoparticles (closed columns), α-gal nanoparticles without anti-Gal (grey columns), or as macrophages alone (open columns, background levels). VEGF secretion by the macrophages was measured by using ELISA in culture media after 24 or 48 h. Data with macrophages from 4 GT-KO mice and their means + S.D. Adapted with permission from [47]. 2018, Elsevier.

**Figure 3 bioengineering-10-01165-f003:**
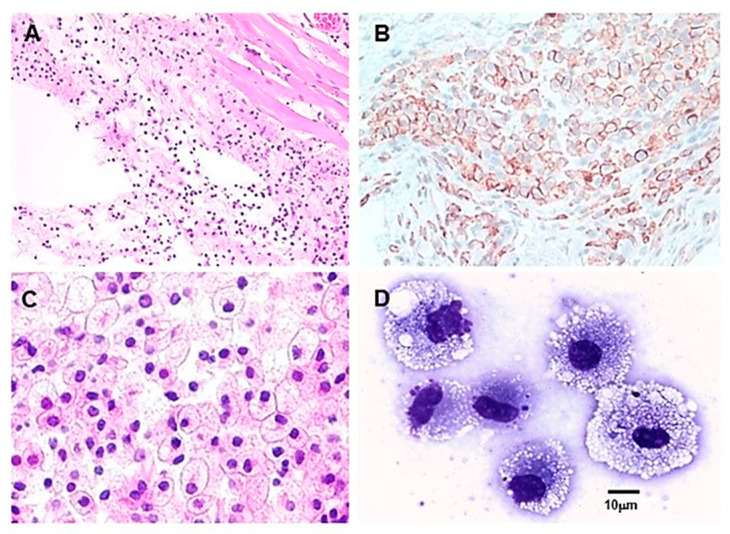
Macrophage recruitment by intradermally injected α-gal nanoparticles (10 mg) in anti-Gal-producing GT-KO mice. (**A**) Macrophage recruitment 24 h following injection of α-gal nanoparticles. The empty oval area is the space formed by the injection of α-gal nanoparticles. The nanoparticles were dissolved by alcohol during staining with hematoxylin and eosin (H&E × 100). (**B**) Macrophages at 4 days post-injection, identified by immunostaining with the F4/80 antibody coupled to peroxidase (HRP) (×200). (**C**) The injection area after 7 days. The site is full of many large macrophages containing vacuoles with morphology characteristic of activated macrophages (H&E × 400). (**D**) Individual macrophages similar to those in (**C**) were harvested from polyvinyl alcohol (PVA) sponge disc containing α-gal nanoparticles. The PVA sponge discs were explanted 6 days post subcutaneous implantation into anti-Gal-producing GT-KO mice. The multiple vacuoles observed in the macrophages are of internalized anti-Gal-coated α-gal nanoparticles (Wright staining, ×1000). Adapted with permission from [47]. Elsevier 2018.

**Figure 4 bioengineering-10-01165-f004:**
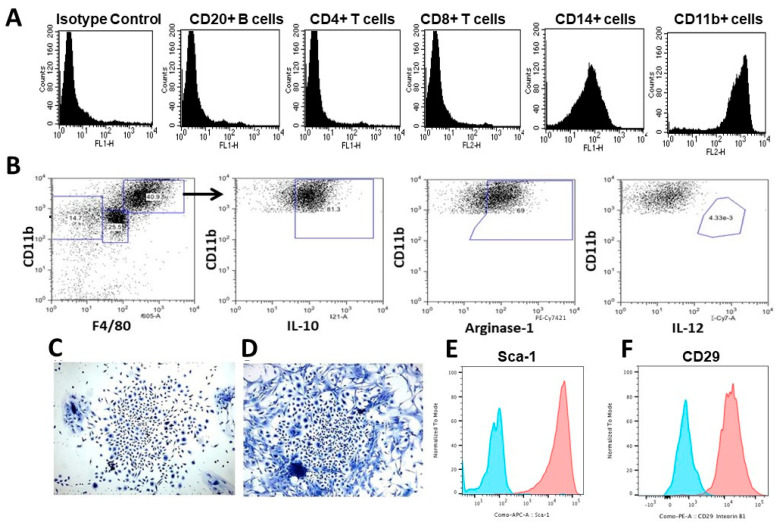
Analysis of cells migrating into PVA sponge discs following anti-Gal/α-gal nanoparticle interaction. (**A**) Flow cytometry analysis of the recruited cells, retrieved from PVA sponge discs containing 10 mg of α-gal nanoparticles, 6 days post subcutaneous implantation. Most of the recruited cells were macrophages expressing CD11b and CD14 cell markers, whereas no significant infiltration of T cells or B cells was observed (representative data of five mice with similar results). (**B**) Analysis of recruited macrophage polarization. The large-size macrophages (CD11b^pos^/F4/80^pos^) were positive also for IL-10 and Arginase-1 but were negative for IL-12. This implied that the majority of the recruited cells were M2 macrophages. (**C**,**D**) Cell colonies formed by what seemed to be MSCs recruited by macrophages migrating into PVA sponge discs containing α-gal nanoparticles and harvested 6 days post subcutaneous implantation. The colonies were observed on day 5 post-culturing. (**E**,**F**) Expression of MSC markers Sca-1 and CD-29, respectively, by cells harvested from colonies like those in (**C**,**D**) (orange curves). Isotype controls are blue curves. Adapted with permission from [57].

**Figure 5 bioengineering-10-01165-f005:**
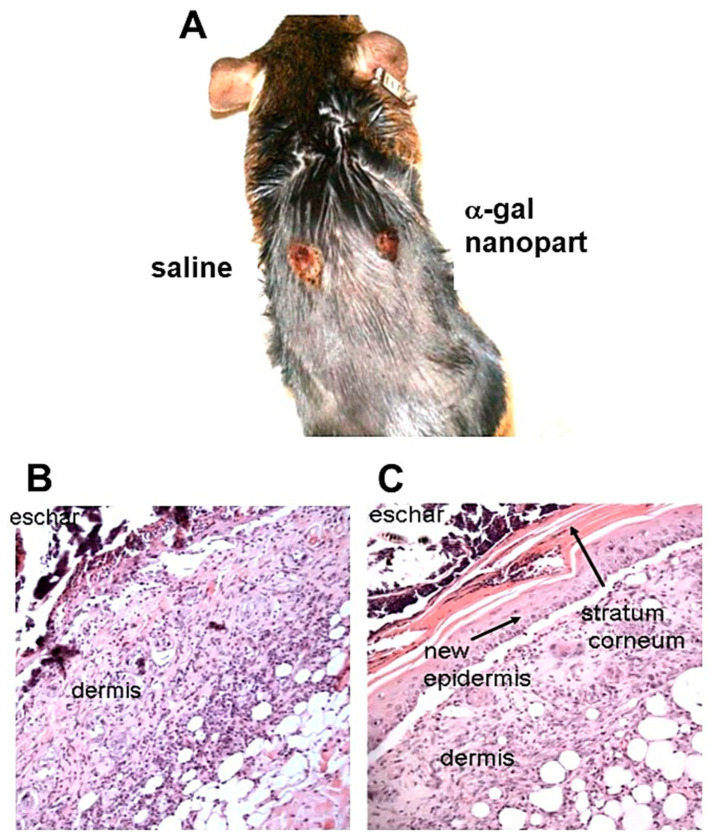
Demonstration of an α-gal nanoparticle-treated burn and saline control burn in a representative anti-Gal-producing GT-KO mouse 6 days post thermal injuries and treatment. (**A**) Gross morphology. Note the big difference in the healing of the two burns. (**B**) Histology of the saline-treated burn presented in A. No epidermis growth is observed over the dermis, which is covered by the eschar. (**C**) Histology of α-gal nanoparticle-treated burn presented in A. The burn is covered by the regenerating epidermis including *stratum corneum,* and the eschar is observed above the regenerating *stratum corneum*. (H&E, ×100). Based with permission on observations from [17]. 2010, Elsevier.

**Figure 6 bioengineering-10-01165-f006:**
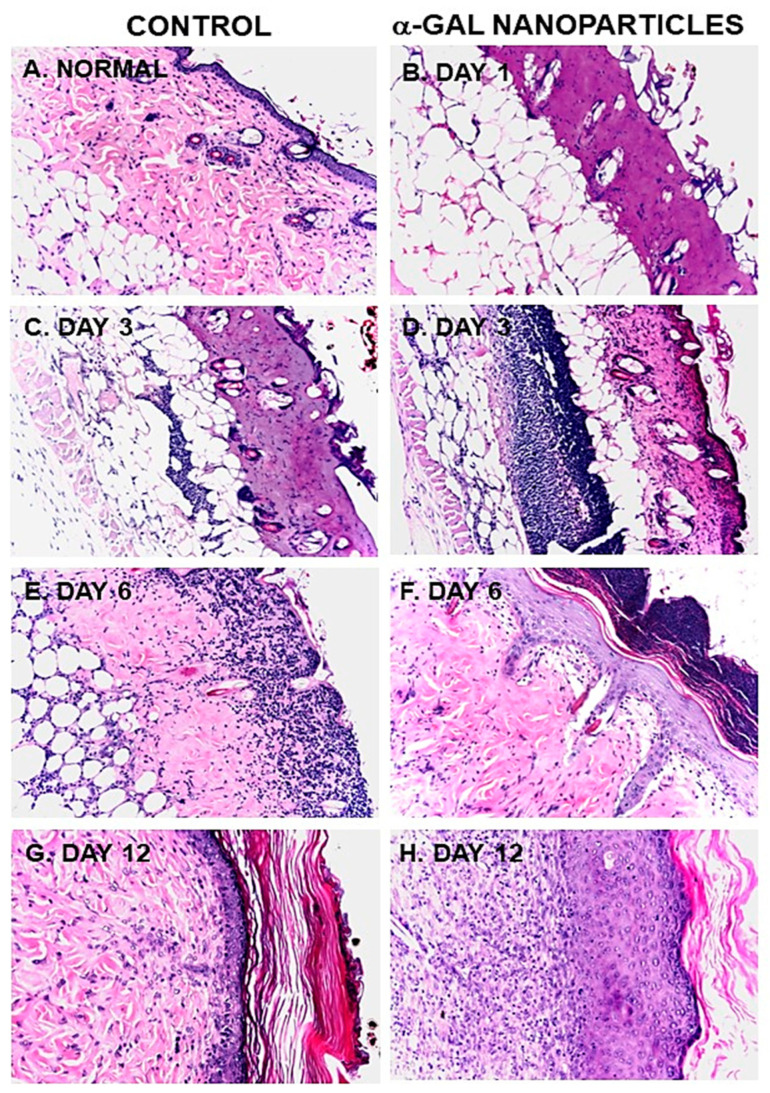
Histology of the accelerated healing of representative burns in anti-Gal-producing GT-KO mice treated with α-gal nanoparticles at various days post-treatment. (**A**) Normal mouse skin. (**B**) A burn, 24 h post injury, displaying histology similar to second-degree burns in humans in that epidermis and dermis are destroyed but not the hypodermis. Saline-treated burns display similar histology. (**C**) Saline-treated burn on day 3. (**D**) α-Gal nanoparticles treated burn on day 3, characterized by extensive recruitment of macrophages and neutrophils in the injured dermis. (**E**) Saline-treated burn on day 6 demonstrating migration of macrophages and neutrophils toward the surface of the burn. (**F**) α-Gal nanoparticle-treated burn on day 6 displaying complete regeneration of the epidermis, including *stratum corneum*. Most of the recruited macrophages demonstrated on day 3 in the dermis are observed on day 6 above and within the apical part of the *stratum corneum*. (**G**,**H**) Day 12 demonstrates complete regrowth of the regenerative epidermis in healing burns treated with saline and α-gal nanoparticles, respectively. With the exception of **A** and **B**, specimens are presented in pairs obtained from the same mouse and are representative of five mice at each time point (H&E, ×100). Reproduced with permission from [17]. 2010, Elsevier.

**Figure 7 bioengineering-10-01165-f007:**
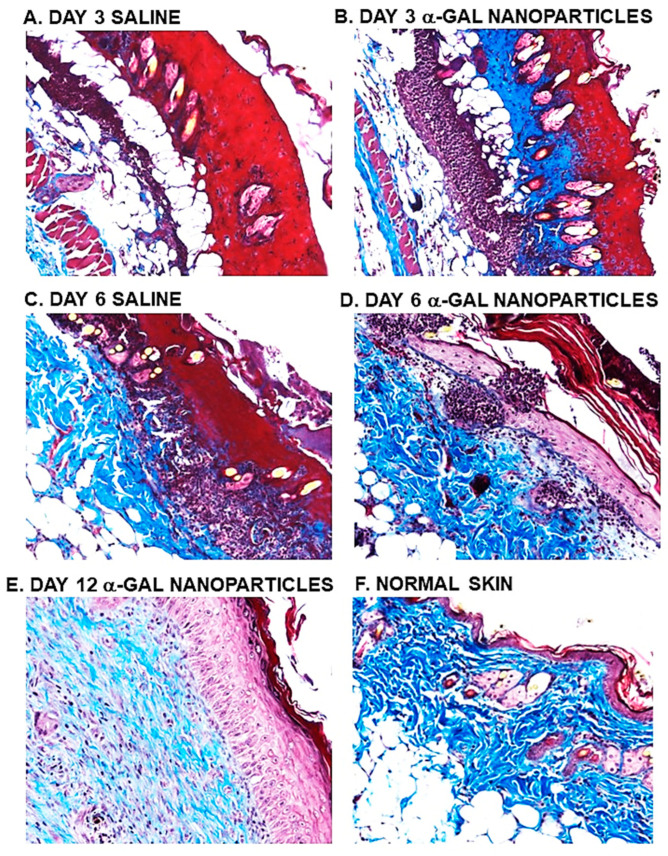
Determination of dermis regeneration as evaluated by Mason trichrome staining blue of de novo formed collagen in saline-treated (**A**,**C**) and in α-gal nanoparticle-treated burns (**B**,**D**,**E**). Normal uninjured skin is presented in (**F**). Specimens (**A**–**D**) are presented in pairs obtained from the same mouse and are representative of five mice at each time point (×100). Reproduced with permission from [17]. 2010, Elsevier.

**Figure 8 bioengineering-10-01165-f008:**
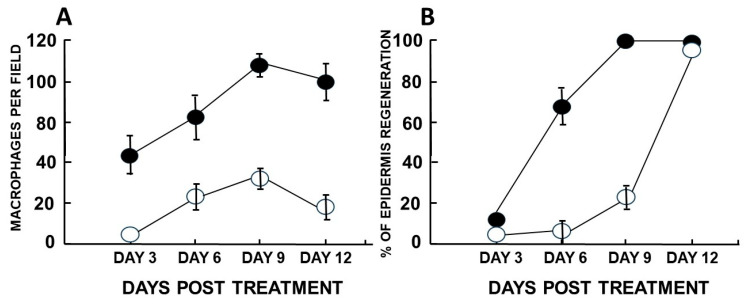
Quantification of macrophage infiltration (**A**) and burn healing determined by % of epidermal regeneration (**B**) in burns treated with α-gal nanoparticles (closed circles) or with saline (open circles). (**A**) The number of infiltrating macrophages at various time points was determined in histological sections by counting cells within a rectangular area marked in a microscope lens at magnification of ×400. (**B**) The proportion (%) of epidermis regeneration was determined histologically by the proportion of the burn surface covered with the newly formed epidermis. Mean ± S.E. from five mice per group. Based on data from [17], with permission. 2010, Elsevier.

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
