# Peer review of "Accelerated Burn Healing in a Mouse Experimental Model Using α-Gal Nanoparticles"

_bioengineering, 2023, doi:10.3390/bioengineering10101165_

Round 1
Reviewer 1 Report
This review/account manuscript summarizes the recent work, mostly by Prof. Galili, on using nanoparticles containing alpha-gal antigen to recruit macrophages to the burn wound, induce the M2 differentiation, and help release growth factors to promote the regeneration of skin cells. Overall, this manuscript showed clear scientific logic and sound experimental data to back that up. The approach of using a glycan antigen is also novel and have huge potential in the therapeutic field. I suggest this manuscript to be accepted in current form.
With that being said, I do have some questions/suggestions if the author have time to answer: 1, I think adding an image of the chemical structure of the alpha-gal antigen would definitely help. 2, Compared to using the alpha-gal nanoparticles to attract antibodies and use the antibodies to attract macrophages, what's the problem with directly applying the antibodies or antibody-conjugated nanoparticles instead of the alpha-gal nanoparticles? 3, During the preparation step, I noticed that it requires the denaturing of proteins. does the alpha-gal antigen also exist in a glycoprotein form? what's the ratio between the glycoprotein form vs the glycolipid form?
Thank you
Author Response
Reviewer 1
- I think adding an image of the chemical structure of the alpha-gal antigen would definitely help.
Response: I thank the reviewer for accepting the manuscript for publication with no requested revisions, except for revising Figure 1. The image of the alpha-gal epitope(Figure 1C) and the corresponding explanation (line 143) were added to Figure 1 and its legend.
- …what's the problem with directly applying the antibodies or antibody-conjugated nanoparticles instead of the alpha-gal nanoparticles?
Response: The rapid requirement of macrophages into the treated burns requires the activation of the complement system by alpha-gal epitopes on the nanoparticles interacting with the natural anti-Gal antibody, like in other antigen/antibody interactions. This complement activation by an antigen/antibody interaction is needed for the production of C5a and C3a complement cleavage peptides which are potent chemotactic factors that direct rapid recruitment of macrophages into the burn (line 113 in the revised manuscript). Administered antibodies or antibody-conjugated nanoparticles cannot activate the complement system without an appropriate antigen. Any such antigen will have to be a self-antigen because no other antigens are present in the burn. Such a self-antigen/antibody interaction may lead to an autoimmune-like activity that could further destroy uninjured cells and tissue surrounding the burn.
- …does the alpha-gal antigen also exist in a glycoprotein form? what's the ratio between the glycoprotein form vs the glycolipid form?
Response: The alpha-gal epitope is also present on glycoproteins because the alpha1,3galactosyltransferase in the trans-Golgi apparatus does not differentiate between carbohydrate chains on glycolipids and glycoproteins. I do not know of studies analyzing the glycoproteins with alpha-gal epitopes on rabbit red cell membranes. We removed the membrane proteins in order to be able to prepare alpha-gal nanoparticles of pure glycolipids, phospholipids and cholesterol. The large amounts of alpha-gal epitopes on glycolipids of rabbit red cell membranes were demonstrated by immunostaining of thin layer chromatography plates using anti-Gal for such staining (see Figure 1 in reference #35 Galili, U.; Clark, M.R.; Shohet, S.B.; Buehler, J.; Macher, B.A. Evolutionary relationship between the anti-Gal antibody and the Galα1-3Gal epitope in primates. Proc. Natl. Acad. Sci. USA 1987, 84, 1369–1373.).
Reviewer 2 Report
The author reviews his current work concerning wound healing induced by anti - Gal antibodies. In this case the focus is on burn healing.
The idea of using antibodies that stimulate the immune system for wound healing sounds very good and the review is is clear and easy to understand, but I have major doubts about its novelty expressed below.
Concerning the review, I noticed that several statements are lacking citations: for example, there are no citations between lines 161 to 172. Though “unpublished observations” is stated in the middle of the paragraph it is unclear which statement it refers to. Please cite or clarify. I also found missing citations between lines 267 and 270 and would like to urge the author to correct the whole review in that respect and not just the two paragraphs mentioned. Moreover, when the text refers to figures that were copied from other work, the figure legends contain the correct citations, but the citations are lacking in the text, so they need to be added to the text.
My major concern is that during the past 3 years the author has written 8 reviews on this topic, as compared to only five original papers. I do not think a ninth review on the same topic can communicate any new idea, so I can not recommend publishing yet another review. I am curious to find new original work on this idea, though.
minor corrections necessary
Author Response
Reviewer 2
- …there are no citations between lines 161 to 172.
Response: Corresponding references were added in lines 162, 167, 171, and 176 of the revised manuscript.
- Though “unpublished observations” is stated in the middle of the paragraph it is unclear which statement it refers to.
Response: The statement “unpublished observations” was revised to ref. 47 in line 171. In addition, the word “This” in line 171 was inserted to clarify the method by which the stability of alpha-gal nanoparticles was evaluated.
- …missing citations between lines 267 and 270
Response: The corresponding reference [48] was added in line 269.
- … urge the author to correct the whole review in that respect (i.e., citations)
Response: Citations have been added and the additions are highlighted.
- ,... the figure legends contain the correct citations, but the citations are lacking in the text, so they need to be added to the text.
Response: Citations included in figures were also added to the corresponding texts and highlighted.
- My major concern is that during the past 3 years the author has written 8 reviews on this topic. I do not think a ninth review on the same topic can communicate any new idea, so I can not recommend publishing yet another review. I am curious to find new original work on this idea, though.
Response: I disagree with this statement. The experimental alpha-gal therapies are applicable (in addition to burn healing) to several clinical disciplines including wound healing, injured heart regeneration post myocardial infarction, regeneration of injured nerves, amplifying viral vaccine immunogenicity, eliciting immune response against autologous tumor antigens, porcine bioimplants in humans, vaccination against protozoa and evolution of primates. As indicated in PubMed, among the 14 publications since 2020, which I authored or coauthored, none of the reviews describe burn healing with alpha-gal nanoparticles, with the exception of “Galili U. Biosynthesis of α-Gal Epitopes (Galα1-3Galβ1-4GlcNAc-R) and Their Unique Potential in Future α-Gal Therapies. Front Mol Biosci. 2021 Nov 4;8:746883”. In that review burns therapy with alpha-gal nanoparticles is discussed in 15 lines out of the whole review which discusses 9 different experimental alpha-gal therapies. In two additional reviews (ref. 58 and “Galili U. Antibody production and tolerance to the α-gal epitope as models for understanding and preventing the immune response to incompatible ABO carbohydrate antigens and for α-gal therapies. Front Mol Biosci. 2023 Jun 28;10:1209974.”), alpha-gal therapy of burns is mentioned in one sentence. Reading the rest of my publications in the last 3 years will indicate that they deal with the other clinical disciplines mentioned above and not with burns therapy.
My motivation for submitting to the special issue of “Tissue Engineering and Regenerative Medicine for Burn Wound Healing” stems from the fact that very few special issues or reviews have dealt in depth with the issue of novel burn therapies. I believe that my submitted review fits this unique special issue, as it provides readers with an opportunity to read in detail the rational and experimental experience with alpha-gal nanoparticles therapy in burns. My hope is that by reading this review, other researchers will be motivated to study this area. I am ready to provide to any researcher interested in this area, mice, antibodies and alpha-gal nanoparticles for performing additional experimental work, in order to further advance this type of therapy.
As for new original works in the recent 3 years, two recent experimental works on alpha-gal nanoparticles therapy were performed in wound healing refs. 57 and 60, which together with ref. 59 validate our wound healing studies in an independent laboratory performing their studies with the alpha-gal nanoparticles described in this review. A third study (ref. 61) describes a similar healing that includes restoration of the original structure and function of the heart in anti-Gal producing GT-KO mice following myocardial infarction (line 448). Presently, we are exploring the possibility of performing Phase I clinical trials in patients with wounds and burns.
Reviewer 3 Report
This review is focused on the use of a-gal nanoparticles for accelerated healing of burns. a-gal nanoparticles exposing a-Gal epitope are prepared from the membranes of erythrocytes of non-primate mammals. Delivery of a-gal nanoparticles to burn wounds together with anti-Gal antibodies results in the activation of complement system and production of chemotactic complement cleavage fragments, which stimulate the migration of macrophages to wound area. Further binding of macrophages to Fc domains of anti-Gal antibodies immobilized on a-gal nanoparticles leads to macrophage activation and production of cytokines and growth factors (e.g. VEGF), which accelerate burn wound healing in animal models. This an excellent review by the specialist who developed the application of a-gal nanoparticles for wound healing. The article is of high interest for a large readership including both biologists and clinicians.
Author Response
Reviewer 3
I thank the reviewer for accepting the manuscript for publication with no requested revisions.
Round 2
Reviewer 2 Report
The author has addressed my concerns in a detailed rebuttal, clearly stating that burn healing had not been addressed in detail in previous reviews. I am convinced by the author's reasoning and so have no concerns about this review any more. From my point of view it can be accepted in its current version.